# Manifestation of Subclinical Extrapulmonary Tuberculosis after COVID-19 Vaccination as Supraclavicular Lymphadenopathy

**DOI:** 10.3390/vaccines10060964

**Published:** 2022-06-16

**Authors:** Han Gyu Cha, Dong Gyu Kim, Joon Ho Choi

**Affiliations:** 1Department of Plastic and Reconstructive Surgery, Soonchunhyang University Bucheon Hospital, Soonchunhyang University College of Medicine, Bucheon 14584, Korea; rlaehdrb2175@gmail.com; 2Department of Nuclear Medicine, Soonchunhyang University Bucheon Hospital, Soonchunhyang University College of Medicine, Bucheon 14584, Korea; 114780@schmc.ac.kr

**Keywords:** lymphadenopathy, COVID-19, vaccination, tuberculosis

## Abstract

Lymphadenopathy after coronavirus disease 2019 (COVID-19) vaccination is a common side effect that usually resolves within several days to weeks, and only observation is recommended. However, for prolonged lymphadenopathy, other possibilities, including malignancy or other lymphoproliferative diseases, may be considered. Herein, we report the case of a 66-year-old woman who experienced prolonged ipsilateral supraclavicular lymph node enlargement after the second dose of the ChAdOx1 (Oxford-AstraZeneca) COVID-19 vaccine, which was eventually diagnosed as extrapulmonary tuberculosis.

## 1. Introduction

The Coronavirus disease 2019 (COVID-19) vaccination program was implemented globally in December 2020 to counteract the pandemic situation of “Severe Acute Respiratory Syndrome-related Coronavirus type 2” (SARS-CoV-2) infection. Although COVID-19 vaccines have proven to lower the morbidity and mortality rate associated with COVID-19, they have the potential to cause side effects. The most common side effect is local inflammation at the injection site followed by enlarged lymphadenopathy which is not a common side effect of other vaccinations. Fortunately, local lymph node (LN) enlargement after vaccination is usually known to resolve in several days to several weeks without sequelae and is recommended to be observed [1,2,3]. However, in cases of persistent LN enlargement, the possibility of cancerous involvement or other lymphoproliferative diseases may be considered. In addition, some infectious diseases that manifest lymphadenopathy should also be considered, especially if fever or respiratory symptoms are combined [4]. It is very rare to diagnose an infectious disease in lymphadenopathy patients without any additional symptoms. Here, for the first time, we reported a case of the prolonged enlargement of the supraclavicular lymph node after COVID-19 vaccination without any other symptoms that resulted in subclinical extrapulmonary tuberculosis (TB).

## 2. Case Description

A 66-year-old female patient visited the outpatient clinic complaining of prolonged ipsilateral supraclavicular LN enlargement that emerged 3 days after she received the second dose of the ChAdOx1 (Oxford-AstraZeneca) vaccine on her left deltoid muscle (Figure 1 and Table 1). She only had injection site tenderness and fatigue after the first vaccination. She was a housewife who had no history of recent traveling or meetings due to the pandemic. She had recovered from acute idiopathic thrombocytopenic purpura 3 years previously and had no other underlying disease. No additional symptoms including fever, cough, myalgia, or weight loss were observed. Vaccine-induced adverse reactions were assumed, and the patient’s condition was carefully monitored. There was no change in the size of the enlarged LN 8 weeks after vaccination without any abnormal laboratory findings (white blood cell count 7830 mm^3^, neutrophil 66.4%, lymphocyte 21.4%, and platelet count 214,000 mm^3^); however, ultrasonography revealed multiple enlarged LNs, with conglomeration in the left supraclavicular area (Figure 2). ^18^F-FDG PET/CT was performed to rule out primary malignancy metastasis. In the results, the ^18^F-FDG PET/CT images revealed multiple hypermetabolic LNs not only in the left supraclavicular area but also in the paratracheal, right hilar, and anterior diaphragmatic areas as well as left gastric, periportal, gastroepiploic, and paraaortic areas. There were no hypermetabolic lesions in both lungs (Figure 3). LN biopsy revealed chronic granulomatous inflammation of the supraclavicular area, with caseous necrosis that was consistent with TB. Real-time polymerase chain reaction (PCR) revealed *Mycobacterium tuberculosis* as the causative agent and the enlarged LN started to diminish after the initiation of rifampin, isoniazid, pyrazinamide, and ethambutol therapy.

## 3. Discussion

Lymphadenopathy is a common transient side effect of COVID-19 vaccination, and LN enlargement ipsilateral to the vaccine injection site has been widely reported [1,5,6]. The sites most involved in palpable lymphadenopathy were reported to be supraclavicular LNs followed by axillary LNs, whereas enlarged LNs were mostly identified in the axilla in non-palpable lymphadenopathy [1]. The average time for clinical resolution was reported to be 7.1 to 26.5 days in previous studies but persistent lymphadenopathy was observed for as long as 10 weeks [7,8,9,10]. Currently, most cases of vaccination-induced lymphadenopathies are recommended to be observed for at least 6 weeks until resolution as they were revealed to be reactive follicular hyperplasia in a cytologic study [11]. To reduce the number of unwarranted biopsies, The National Comprehensive Cancer Network recommends delaying imaging studies by 4 to 6 weeks following COVID-19 vaccination [12]. Furthermore, a recent prospective study on the US evaluation of COVID-19 vaccine-associated lymphadenopathy concluded that short-term follow-up within 6 weeks is not helpful and at least 12 weeks after vaccination may be reasonable for the resolution of lymphadenopathy [13].

However, there are some reports regarding persistent LN enlargement that eventually have been diagnosed as pathological diseases including Kikuchi-Fujimoto disease, hemophagocytic lymphohistiocytosis, and sarcoidosis [14,15,16]. Furthermore, while flares of disease in known immune-mediated disease patients following vaccination have been detected, vaccine-associated immune-mediated diseases have also been reported in healthy subjects after vaccination [17]. In addition, we cannot ignore the possibility of malignancies such as lymphoma because COVID-19 vaccination-induced lymphadenopathy has been reported to mimic the progression of oncological patients [18,19]. The most uncertain situation is when facing an asymptomatic patient with persistent LN enlargement alone. Differential diagnosis should be performed among not only autoimmune diseases and malignancies but also in terms of infectious diseases including infectious mononucleosis, toxoplasmosis, and other viral infections (cytomegalovirus, herpesvirus, and human immunodeficiency virus) [4,20]. Lastly, the possibility of mycobacterial or fungal infections should not be underrated.

Extrapulmonary TB affects sites other than lungs, such as pleura, LNs, bones and joints, the genitourinary tract and meninges, and the involvement of LNs is most common. It is reported to affect 18–28% of all TB cases but is frequently diagnosed late due to its wide variety of clinical manifestations and heterogeneous nature [21,22]. Although the development of new imaging tools including ^18^F-FDG PET/CT, which were used in this case, have enabled earlier detection, it is frequently excluded in differential diagnosis especially in asymptomatic patients who are classified as subclinical TB. The World Health Organization aims to diagnose these patients before they become symptomatic or infectious as they are predicted to reactivate in 5 to 10% of people [23,24]. However, the complete blood count including white blood cell counts, platelet counts and hemoglobin level is generally normal in subclinical TB patients. Inflammatory markers, the erythrocyte sedimentation rate and C-reactive protein, are also known to be in the normal range.

Ultrasonography may be the most simple and non-invasive diagnostic tool in cases of extrapulmonary TB with LN enlargement. Typical sonographic features of TB lymphadenitis are multiple, irregular, enlarged, and conglomerating lymph nodes that have a tendency toward fusion [25], which is the exact finding in our case. ^18^F-FDG PET/CT can also contribute to earlier extrapulmonary TB detection particularly in multisite involvement cases with an average maximum standardized uptake value (SUVmax) of 5.3–13.4 [26,27]. Multiple hypermetabolism has an SUVmax of 5.1–9.5 in cases where the patient presents a typical characteristic of extrapulmonary TB in ^18^F-FDG PET/CT. Another simple option prior to biopsy for screening would be the tuberculin skin tests, for which positive results should be further confirmed by in vitro enzyme immunoassay tests [28].

To our knowledge, this is first report of subclinical TB manifestation after COVID-19 vaccination. We cannot rule out the possibility of a coincident occurrence of extrapulmonary TB in this endemic region. Nevertheless, the patient had no other symptoms including cough, fever, and weight loss, other than LN enlargement immediately after the vaccination. We suspect that the immunologic reaction to vaccination triggered the manifestation of subclinical TB which resulted in lymphadenopathy. A previous study reported that subclinical TB and COVID-19 infection show a similarly increased abundance of circulating myeloid subpopulations and this shared pathway of immunopathogenesis suggests that COVID-19 infection triggers the progression of TB [29]. Similarly, the ChAdOx1 vaccine, an adenoviral vectored vaccine with a full length SARS-CoV-2 spike insert may potentially trigger subclinical TB as it acts as an inflammatory adjuvant and is assumed to be related to innate and adaptive immune mechanisms. The factors that are known to control the latent infection include macrophage, CD4 T cells, CD 8 T cells, and gamma interferon (IFN-γ) [30]. However, the percentages of the IFN-γ and CD56 natural killer (NK) cell have been proven to reduce in elderly patients after their first ChAdOx1 vaccination and restore following their second vaccination owing to immunosenescence [31]. As NK cells play a critical role in immunity against tuberculosis [32], the enlargement of regional draining LNs after second ChAdOx1 vaccination in our 66-year-old patient may be related to the recruitment of NK cells and the production of IFN-γ. A recent study also described nodal reactivity as being more common after the second dose of COVID-19 vaccination [33]. Therefore, future research on the relationship between subclinical TB and post-vaccination lymphadenopathy should be focused on role of NK cells and the production of IFN-γ.

In conclusion, in cases of prolonged COVD-19 vaccination-induced lymphadenopathy, pathologic states including not only malignancy but benign diseases such as TB should be considered as well, especially in endemic regions. In asymptomatic lymphadenopathy patients, ultrasonography or PET/CT is a valuable tool for early evaluation, differential diagnosis, and eventually for early treatment.

## Figures and Tables

**Figure 1 vaccines-10-00964-f001:**
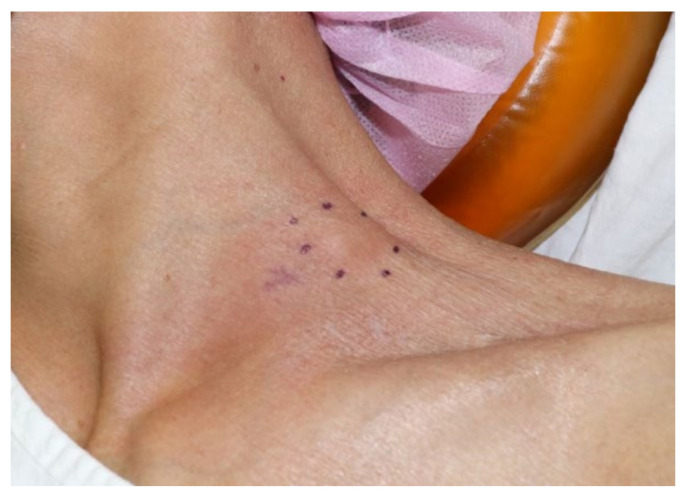
Clinical photograph of a 66-year-old female patient with enlarged lymph node in left supraclavicular region ipsilateral to the COVID-19 vaccine injection site.

**Figure 2 vaccines-10-00964-f002:**
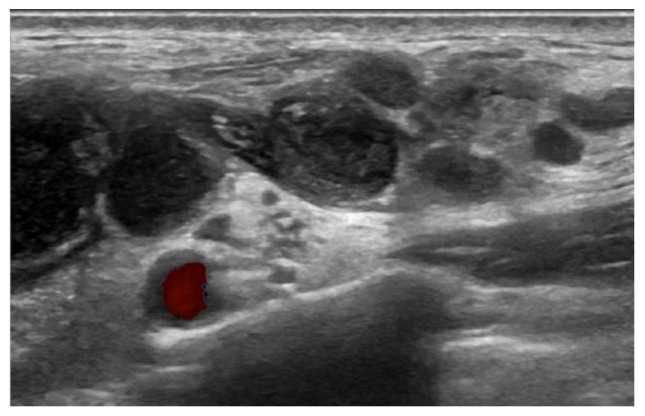
Ultrasonography scan of COVID-19 vaccination-induced lymphadenopathy patient revealed multiple enlarged LNs, with conglomeration in the left supraclavicular area.

**Figure 3 vaccines-10-00964-f003:**
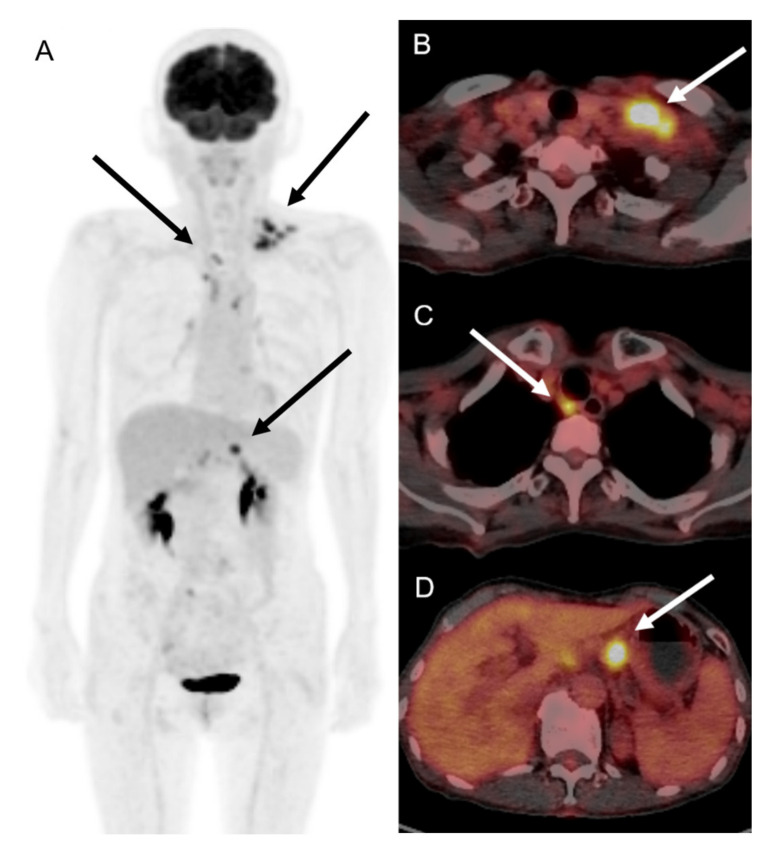
^18^F-FDG PET/CT images of patient with COVID-19 vaccination-induced lymphadenopathy patient. Maximum intensity projection (**A**) and tomographic axial images of ^18^F-FDG PET/CT showed multiple hypermetabolism in left supraclavicular (**B**), retrotracheal (**C**), and left gastric artery lymph nodes (**D**), with maximum standardized uptake value of 5.1–9.5.

**Table 1 vaccines-10-00964-t001:** Side effects after 1st and 2nd dose of ChAdOx1 COVID-19 vaccine in case patient.

	1st Dose	2nd Dose
Symptoms	Fatigue	Lymphadenopathy
	Injection site tenderness	

## Data Availability

Not applicable.

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
