# Peer review of "Manifestation of Subclinical Extrapulmonary Tuberculosis after COVID-19 Vaccination as Supraclavicular Lymphadenopathy"

_vaccines, 2022, doi:10.3390/vaccines10060964_

Round 1
Reviewer 1 Report
Dear Authors
The manuscript is very interesting as it deals with one of the consequences of the vaccine for COVID-19.
However, I think that the literature review can be improved as well as the method and the results.
My Best Regards
Author Response
Thank you for your precious suggestion. We have added some more literature reviews in discussion section to discuss more about the relationship between COVID-19 vaccination and activation of subclinical TB.
We also added a table describing the side effects after 1st and 2nd dose of Covid-19 vaccine in this case patient with laboratory findings.
Reviewer 2 Report
In the current manuscript, Choi et al. showed a case report of a 66-year-old female who received the second dose of Oxford-AstraZeneca COVID-19 vaccine and had extrapulmonary tuberculosis. The authors concluded that the COVID-19 vaccine could cause rare TB incidences in TB endemic regions. Overall, I think the author's conclusions do not stand scientifically. TB is a chronic disease and for it to be extrapulmonary, the patient had to be infected with Mtb for months. The time authors discuss from being inoculated to having a clinical TB diagnosis does not provide any evidence to conclude that the TB might have resulted from the COVID-19 vaccine.
Author Response
Thank you very much for your comment. However, I think there is some misunderstanding regarding to our conclusion in that we are reporting that COVID-19 vaccine has triggered the manifestation of extrapulmonary TB that has been in such latent phase. We do not think that COVID-19 vaccination is the cause of TB but may have role on manifesting TB by immunologic effect. Therefore, we have added some more literature reviews to discuss more about the relationship between COVID-19 vaccination and activation of TB.
Reviewer 3 Report
Known in the field based on previous literatures:
- Covid-19 is an infectious disease caused by severe acute respiratory syndrome coronavirus 2 (SARS-CoV-2). The disease has blowout worldwide, leading to an ongoing pandemic.
- Symptoms of COVID‑19 are variable, but often include cough, breathing difficulties, headache, fever, fatigue, loss of smell and taste.
- COVID‑19 vaccine is developed and intended to provide acquired immunity against the virus.
In this case report authors reported following observation:
- Although, Covid-19 vaccine is effective, but several side effects were speculated. In this manuscript, authors reported another side effect in the case of a woman who had prolonged supraclavicular lymph node enlargement after the 2nd dose of the Oxford-AstraZeneca COVID-19 vaccine, which was ultimately diagnosed as extrapulmonary tuberculosis.
- In this case study, authors reported lymphadenopathy (swelling of lymph nodes which could be also due to bacterial, viral, or fungal infections, and malignancy) after 3 days of the second dose of the vaccination.
- Authors performed PET/CT to rule out primary malignancy and RT-PCR to confirm extrapulmonary tuberculosis.
The data provided by authors based on a single case which is insufficient to say side effect of vaccine. The following suggestions if incorporated could help in the better understanding of the significance of the study.
Minor Concerns:
- Please add references at the end of the first sentence of discussion- Lymphadenopathy is a common transient side effect of COVID-19 vaccination, and LN enlargement ipsilateral to the vaccine injection site has been widely reported (add references here).
- Why have you not measured the inflammatory cytokines in this case? Explain, how could you suspect immunologic reaction to vaccination triggered the TB which resulted in lymphadenopathy?
- What was the similarity and difference of side effects after 1st and 2nd dose of Covid-19 vaccine in this case ? Add a table of side effects of Covid-19 and compare it to TB.
Author Response
1. Please add references at the end of the first sentence of discussion- Lymphadenopathy is a common transient side effect of COVID-19 vaccination, and LN enlargement ipsilateral to the vaccine injection site has been widely reported (add references here).
Thank you very much for the comments. We have added references at the end of the first sentence of discussion- Lymphadenopathy is a common transient side effect of COVID-19 vaccination, and LN enlargement ipsilateral to the vaccine injection site has been widely reported [1,4,5].
2. Why have you not measured the inflammatory cytokines in this case?
We have not measured the inflammatory cytokines but rather measured inflammatory markers including WBC, neutrophil, lymphocyte, and platelet count and have added the results in the manuscript.
Explain, how could you suspect immunologic reaction to vaccination triggered the TB which resulted in lymphadenopathy?
We have added our discussion on reason for triggering.
"The factors that are known to control the latent infection include macrophage, CD4 T cells, CD 8 T cells, and gamma interferon (IFN-γ) [21]. However, the percentages of IFN-γ and CD56 natural killer (NK) cell have been proven to reduce in old patients after first ChAdOx1 vaccination and restore in second vaccination owing to immunosenescence [22]. As NK cells play a critical role in immunity against tuberculosis [23], the enlargement of ipsilateral regional draining LNs after second ChAdOx1 vaccination in our 66-year-old patient may be related to recruiting of NK cells and production of IFN-γ as a consequence of immunosenescence."
3. What was the similarity and difference of side effects after 1st and 2nd dose of Covid-19 vaccine in this case ? Add a table of side effects of Covid-19 and compare it to TB
We have added a table for <Side effects after 1st and 2nd dose of ChAdOx1 COVID-19 vaccine in case patient> and described that patient had none of symptoms of TB including cough, fever, and weight loss.
Round 2
Reviewer 2 Report
The authors have addressed my major concerns by adding additional commentary to the discussion. The manuscript is now suitable for publication.
Author Response
Thank you very much.